# Steroidal Antimetabolites Protect Mice against *Trypanosoma brucei*

**DOI:** 10.3390/molecules27134088

**Published:** 2022-06-25

**Authors:** Minu Chaudhuri, Ujjal K. Singha, Boden H. Vanderloop, Anuj Tripathi, W. David Nes

**Affiliations:** 1Department of Microbiology, Immunology, and Physiology, Meharry Medical College, Nashville, TN 37208, USA; ujjal.singha@vumc.org (U.K.S.); anuj1tripathi@yahoo.com (A.T.); 2Department of Chemistry & Biochemistry, Texas Tech University, Lubbock, TX 79409, USA; boden.h.vanderloop@vanderbilt.edu

**Keywords:** ergosta-5,7,22,24(28)-tetraenol (ERGT), cholesta-5,7,22,24-tetraenol (CHT), ergosterol biosynthesis, antimetabolite, suicide substrate, *Trypanosoma brucei*

## Abstract

*Trypanosoma brucei*, the causative agent for human African trypanosomiasis, is an emerging ergosterol-dependent parasite that produces chokepoint enzymes, sterol methyltransferases (SMT), not synthesized in their animal hosts that can regulate cell viability. Here, we report the lethal effects of two recently described natural product antimetabolites that disrupt *Acanthamoeba* sterol methylation and growth, cholesta-5,7,22,24-tetraenol (CHT) and ergosta-5,7,22,24(28)-tetraenol (ERGT) that can equally target *T. brucei*. We found that CHT/ERGT inhibited cell growth in vitro, yielding EC_50_ values in the low nanomolar range with washout experiments showing cidal activity against the bloodstream form, consistent with their predicted mode of suicide inhibition on SMT activity and ergosterol production. Antimetabolite treatment generated altered *T. brucei* cell morphology and death rapidly within hours. Notably, in vivo ERGT/CHT protected mice infected with *T. brucei*, doubling their survival time following daily treatment for 8–10 days at 50 mg/kg or 100 mg/kg. The current study demonstrates a new class of lead antibiotics, in the form of common fungal sterols, for antitrypanosomal drug development.

## 1. Introduction

*Trypanosoma brucei*, a group of flagellated parasitic protozoa, infect both human and domestic animals and cause a fatal disease, African trypanosomiasis (AT) [1]. The disease is transmitted by an insect vector, the tsetse fly. During its digenetic life cycle, *T. brucei* undergoes a complex developmental process [2]. The procyclic and the bloodstream forms are the two major developmental forms of *T. brucei* that are found in the insect gut and the mammalian blood, respectively. World Health Organization (WHO) classified AT as a neglected tropical disease. The available drugs for AT originate as synthetic compounds, are mostly toxic, antiquated, develop resistance and difficult to administer [3]. Recent efforts of the Drugs for Neglected Diseases Initiative (DNDi) developed two new promising oral drugs, fexinidazole and oxaboroles, by screening small molecule chemical libraries against *T. brucei* [4,5]. Despite their promiscuity, fexinidazole has potential to develop cross-resistance to nifurtimox [6], a widely used current therapy for AT, and the actual target for oxaboroles is yet to be identified. Interestingly, an alternative approach that has recently advanced new trypanocides against AT derives from the isolation and testing natural products through traditional bioactivity-guided isolation steps or from the recently applied genome mining and engineering efforts [7,8]. Therefore, these successes involving synthetic compounds or natural products notwithstanding, there continues to be an urgent need to develop new anti-parasitic drugs that can prevent parasite growth through novel modes of action that might include blockage of crucial enzymes in essential metabolic pathways in African trypanosomes [9,10].

Antimetabolites in primary metabolism have emerged as effective chemotherapeutic agents against some forms of cancer by selectively disrupting a productive biosynthetic step [11,12]. Typically, these compounds are designed to show substrate mimicry and under physiological conditions they are expected to enter cells, then outcompete a relevant intermediate for a targeted enzyme that provides an essential end-product. Most of these antimetabolites are synthetic compounds with a functional warhead, making them suicide substrates. Consequently, the inhibitory influence of these drugs goes beyond simple competitive inhibition; following irreversible binding of inhibitor via covalent attachment to the active site of the enzyme causing its inactivation accompanied by muted biosynthesis of the essential end-product that, therefore, produces suppression of cell growth and development [13,14,15,16]. Covalent drugs have gained further notoriety [17,18] in their use to treat AT. Thus, a highly effective fluorinated substrate analog was developed in the later part of the 20th century against the ornithine decarboxylase in the polyamine biosynthesis pathway and appears under the brand name, Ornidyl, among others, and its chemical name, eflornithine [9].

Noting recent therapeutic successes involving other small molecule inhibitors, VNI targeting 14α-sterol demethylase enzyme in the trypanosome ergosterol biosynthesis pathway [19], we considered the sterol methyltransferase (SMT) enzyme as another druggable target for ergosterol biosynthesis in trypanosomes. SMTs have been shown to form covalent attachments to their substrate analogs, similar structurally to zymosterol (fungi/protozoa) or cycloartenol (plants), making these analogs suicide substrates (Figure 1) [20]. In contradistinction to the mechanism of action of typical ergosterol biosynthesis inhibitors, such as VNI, which bind through electrostatic interactions, the substrate mimics we designed produce their electrophilic handle only after the catalytic reaction cycle is turned on, yielding a sterol methyl intermediate positively charged in an unconventional side chain location (Appendix A), otherwise these analogs are functionally inert upon binding. So far as is known, inhibitors coupled to cell death that interfere with ergosterol production and processing typically generate altered plasma-membrane structures, or in some cases altered mitochondrial membranes, as a consequence of changes to apoptosis-like and autophagic events [21,22]. Fascinated by the possibility that suicide inhibitors could interfere with sterol methylation in *Acanthamoeba* [22], it was thought that they could be developed elsewhere to control the ergosterol-dependent disease processes associated with parasites. Thus, we initiated a program several years ago to identify suicide inhibitors (synthetic) that specially block the sterol methylation pathways of *T. brucei* (*Tb*) or *Acanthamoeba castellanii* (*Ac*) and inhibit the ergosterol biosynthesis pathways in these protozoa without effect on cholesterogenesis in animals [23,24,25,26]. Our work is based on the condition that the major genetic difference in host C_27_-cholesterol biosynthesis pathway and parasites *Tb* and *Ac* C_28_-ergosterol and C_29_-7-dehydroporifersterol biosynthesis pathways [22,24] (Figure 1) reside in the sterol methyltransferase (SMT) gene synthesized in *Ac* and *Tb* but not in animals [27]. Because the variant SMTs in these pathogens are product specific and mechanistically distinct in kinetoplastids and amoebae yielding Δ^24(28)^- or Δ^25(27)^-24-alkyl sterols (Appendix A), the differences in the reaction pathway of sterol methylation in these organisms and their absence in the host organism affords a path to drug selectivity in the form of our newly discovered steroidal chemotherapeutics (synthetic compound) and steroidal antibiotics (natural compound) that uniquely target SMT activities.

In our search for more effective synthetic mimics to complex SMT, we surreptitiously observed a pair of natural substrates, cholesta-5,7,22,24-teraenol (CHT) and ergosta-5,7,22,24(28)-tetraenol (ERGT), produced in the yeast ergosterol biosynthesis pathway that could bind and alter catalysis of either the *Ac* or *Tb* SMT [26,27,28,29] in similar fashion to that of 26,27-dehydrozymosterol tested against the yeast SMT [30,31]. For these reasons, we asked whether CHT or ERGT could bind *Tb* SMT in vivo and in so doing, play a protective role in animals infected with the parasite. We chose to study mice infected with *Tb* since several reports revealed the ergosterol biosynthesis pathway can be uncoupled, producing cell death, when the cells are subjected to inhibitors targeting crucial enzymes in the post-squalene pathways of trypanosome steroidogenesis [10,32]. Consistent with these reports, in our studies of the transition state inhibitor 25-azalanosterol against *Tb* SMT, we observed the analog prevented Tb cell proliferation in the low micromolar range [33] and quelled the parasite burden in vivo, effectively protecting the infected mice from the disease [25]. Because the recently discovered steroidal antimetabolites CHT and ERGT, distinct from the conventional anti-infectives of ergosterol synthesis [21,34], can covalently bind the *Tb* SMT during catalysis [30], this predicts their suicide inhibitor properties in *Tb* ergosterol biosynthesis and in so doing, provides a new armamentarium for therapeutic study. Here, mice infected with *Tb* parasites were treated with the suicide inhibitors CHT and ERGT or with no inhibitor. The effects of these short-term treatments upon cell proliferation and subcellular organization in vitro and in vivo were determined.

## 2. Results

### 2.1. ERGT/CHT Generates Marked Inhibition of T. brucei BF Growth in Cell Culture

From our recent observation that a sparking level of ergosterol, i.e., hormonal/trace amount of 24-alkyl sterol produced biosynthetically in bloodstream form (BF) [25] against bulk amounts of cellular cholesterol derived from the culture medium [35], is necessary for trypanosome growth and the report showing in *Trypanosoma cruzi* that sterol methylation is also essential in vivo and in vitro for trypanosome growth [36], we were encouraged to look for new leads, in the form of CHT and ERGT, that could be used to treat pathogens capable of catalyzing the sterol methylation route yielding a D25(27)-product. As observed previously, certain intermediates of the yeast sterol biosynthesis pathway, cholesta-5,7,22,24-tetraenol (CHT) and ergosta-5,7,22,24(28)-tetraenol (ERGT), while ineffective against the *Ac* SMT1 enzyme, which generates a Δ^24(28)^-product, can nonetheless inhibit sterol methylation catalyzed by other SMTs such as the *Ac* SMT2 isoform capable of generating an alternate Δ^25(27)^-product. Importantly, incubation of these fungal sterols at the low nanomolar range against *Ac* trophozoites cells led the substrate to distinctly bind *Ac* SMT2, followed by cellular loss of 24-alkyl sterol and cell death within hours of the initial cell treatment [28]. Strikingly, as determined for the related suicide inhibitors previously studied, the depleting effect of the foreign sterol containing a reactive conjugated diene on ergostanoid biosynthesis was specific to the pathogen while ineffective against the cholesterol biosynthesis pathway in cultured HEK cells, which did accumulate CHT and ERGT and metabolize them to Δ^5,22,24^-trienol products [28].

Noting the similarity in sterol methylation patterning catalyzed by *Ac* SMT2 and *Tb* SMT1 in generating a Δ^25(27)^-product (Appendix A) [28,29] and the understanding that *Tb* cell growth is vulnerable to a loss of sterol methylation synthesis, we proceeded to test CHT and ERGT on *Tb* cell growth in culture. Preliminary tests of these compounds yielded potent growth inhibition of bloodstream form (BF) cells in cell culture, relevant to any future studies of their efficacy in animal models of infection. Consistent with the *Ac* studies [28], the BF is highly sensitive to both CHT and ERGT (Figure 2), displaying calculated EC_50_ values of 2.9 ± 0.2 nM and 0.52 ± 0.05 µM for CHT and ERGT, respectively, while the EC_90_ for these compounds was 8.8 ± 0.5 nM and 2 ± 0.3 µM, respectively. We have shown previously that CHT and ERGT do not have any effect on growth of the human epithelial kidney cells (HEK) in culture when treated with up to 40 µM concentrations [28] of either CHT or ERGT. Consequently, the approximate specificity index of CHT and ERGT calculated against the cell growth of HEK and *Tb*-BF is ~10^3^, showing targeted potency toward trypanosomes and not the human host.

### 2.2. CHT and ERGT Cause Cell Death within Hours of Treatment

Microscopically evident, altered morphology associated with death was detected when the bloodstream form (BF) cells were incubated at EC_90_ of either ERGT or CHT. To investigate this process further, we incubated cells (1 × 10^6^/mL) at 5 × EC_50_ concentrations of CHT and ERGT separately for a shorter period (2–4 h) and observed the cell morphology under the microscope (Figure 3A–C). During these conditions cell viability was minimally affected. Cells treated with vehicle control were compared in parallel. The control cells were intact, fully motile, and alive at both time points (2 and 4 h), as expected (Figure 3A). In contrast, treated cells acted differently. At the 2 h time point, BF cells treated with either CHT or ERGT were mostly intact and motile but some cells (1–2%) tended to circularize (Figure 3B,C, left panel). However, at the 4 h time point almost 90% of cells were circularized, flagella detached, and appeared dead (Figure 3B,C, right panel). Therefore, CHT/ERGT kills *Tb*-BF very rapidly, like *Ac*. The effect of CHT on *Tb*-BF cell morphology was more drastic in comparison to ERGT.

### 2.3. Antimetabolites Cause Mitochondrial Swelling and Mitophagy 

To observe if there were any changes in the intracellular structure of *T. brucei* due to the treatment with antimetabolites, we harvested BF cells after 4 h treatment with a representative antimetabolite, ERGT, at 5 × EC_50_ concentration and prepared them for electron microscopy (EM) (Figure 4A,B). Captured images revealed many vesicular structures inside the treated cells in comparison with control (Figure 4B). We found the nucleus (N) and kinetoplast (K) DNA, the mitochondrial DNA in trypanosomes [37] in the control cells as expected (Figure 4A). The presence of dividing nucleus and extended kinetoplast in the control indicated that the cells were replicating. In contrast, in the ERGT-treated cells we did not identify any dividing nucleus, instead the presence of autophagosomes (Figure 4B) was seen. In contrast to the tubular mitochondria in the control (Figure 4A), mitochondria in the ERGT-treated cells were enlarged without any cristae structures (Figure 4B). These results show a significant alteration in the intracellular structure, including a major change in the mitochondria in the BF that may lead to cell death upon treatment with these antimetabolites.

### 2.4. Washout Experiments of CHT and ERGT Reveal Inhibition of BF Growth and by Analogy Ergosterol Production

We investigated to determine whether washout experiments in the form of an extended duration of action can be observed through growth response. Here, we assumed that covalent adduct of antimetabolite SMT should remain upon seeding treated cells into fresh medium that thereby prevents cell growth, otherwise the steroidal antimetabolite should undergo conversion to product, freeing up SMT for new catalysis that therefore permits a resumption in ergosterol production and cell growth to the control levels. For this purpose, the BF cells were treated with CHT and ERGT at respective 5 × EC_50_ concentrations for 2 h. Cell numbers were not decreased during this short-term treatment and cells were mostly intact and motile as shown in Figure 3B,C (left panels). Cells were harvested, washed to remove any residual compounds and reinoculated in fresh medium with different cell numbers (10^4^/mL, 10^3^/mL, 10^2^/mL, and 10^1^/mL) and allowed to grow in appropriate conditions for 4 days, the point when the BF cells reached the stationary phase when inoculated at 10^3^/mL (Figure 5A). Once reaching the stationary phase, BF cells die very rapidly due to a quorum-sensing-like phenomenon [38]. Therefore, control cells inoculated at 10^4^/mL reached at the maximum level before day 4 and some cells died thereafter (Figure 5A), as expected. In contrast, we found that BF cells treated with either CHT or ERGT had significant growth inhibition in comparison with control after removal of the antimetabolites. Cell growth was almost undetectable till day 4 after washout of the CHT (Figure 5B). This indicates that CHT treatment caused an inherent damage in the BF, which was not recovered even after removal of the antimetabolites. Cells treated with ERGT also showed a lingering growth inhibitory effect (Figure 5C). Cell numbers were below the detection levels up to 3 days, even when inoculated at 10^4^/mL or 10^3^/mL. After that, cells grew slowly and reached the level of 2 × 10^6^/mL at day 4. Therefore, as observed for inhibitor sensitivity, treatment with CHT had longer effects on BF cell growth in comparison with ERGT. These results strongly support our previous observation that these antimetabolites covalently bound and inactivate the target enzyme SMT. Moreover, these different antimetabolite treatment outcomes are to be expected since the ERGT/CHT inhibited growth response is subject to endogenous *Tb* SMT1 acceptance of the fungal sterols. As reported, a first methylation substrate (e.g., CHT) is converted by the *Tb* enzyme about ten times more effectively than a second methylation substrate (e.g., ERGT) [23].

### 2.5. Protective Role of CHT/ERGT on Mice Infected with T. brucei 

To test the efficacy of CHT and ERGT in *T. brucei* infection, groups of mice were infected with the *T. brucei* 427 strain, which creates an acute form of the disease. Since we had some success with treating mice infected with *Tb* at doses of 5 mg/kg 25-azalanosterol [25], we tested CHT or ERGT at this dosage in the *Tb*-infected mouse with no beneficial results. Consequently, we then tested a higher level of steroidal antimetabolite to outcompete the circulating cholesterol for lipoprotein binding necessary for uptake by the *Tb* bloodstream form [35]. For these studies, mice were divided into six groups (five mice in each group). Starting at the same day of infection, group-I and -II mice were treated with ERGT (50 mg/kg and 100 mg/kg body weight, respectively) each day. The group-III and -IV infected mice were similarly treated with CHT (50 mg/kg and 100 mg/kg body weight, respectively) each day and the group-V and -VI were vehicle control for 50 mg/kg and 100 mg/kg dosages, respectively. As expected, the control infected mice had all died by 5–6 days post-infection, whereas the treated mice lived longer. In further detail, three of the five ERGT-treated (50 mg/kg body weight) mice died on day 8 and rest had died by 10 days post-infection (DPI) (Figure 6A). We didn’t observe any better results when increasing the dose to 100 mg/kg/day (Figure 6B). The results show that like in in vitro data, CHT has a slightly higher trypanocidal efficiency than ERGT, consistent with their substrate recognition. However, doubling the dosages didn’t show much additive effect. This is possibly because the compounds are metabolized at the sterol core structure [28] by the host. Counting the parasite numbers in the blood of the infected mice revealed that the parasite numbers were below the detection levels (~10^3^/mL) in the first 2 days of post-infection, increased exponentially after that period, and reached maximum by day 5 in the control groups (Figure 6C,D), which are the characteristics of the acute form of the disease in laboratory animals [39,40]. In contrast to the control, we couldn’t detect any parasites in the blood till 6 days post-infection, when treated with ERGT (50 mg/kg/day) (Figure 6C). The blood parasitemia reached maximum levels at different time points and most of the mice stayed alive till day 8. Blood parasite counts were below the detection level up to day 7 in the group of mice treated with CHT (50 mg/kg body weight/day); however, after that it increased exponentially (Figure 6C). Increasing the doses of ERGT or CHT (100 mg/kg/day) didn’t show any further delay in the appearance of the parasite number in the blood above the detection limit (Figure 6D). Neither did it take any longer time to reach lethal levels. Overall, our studies showed that ERGT/CHT protected mice infected with *T. brucei* by doubling their survival time, which is equivalent to 100% increase in life expectancy, following daily treatment for 8–10 days at 50 mg/kg.

## 3. Discussion

Fungi, plants, and protozoan species use SMT for generation of a vast array of 24-alkyl sterol structures necessary for cell growth and development [41,42]. The primary sequences and tetrameric subunit organization of a wide range of these catalysts reflect a common active-site topography that can accept a wide range of structurally similar substrates. However, under selection pressure, this enzyme evolved to accept alternate substrate side chain groups enabling species-specific product formation and an SMT-specific dead-end complex [43,44,45]. As determined, it is straight-forward to detect the isolated CHT-SMT complex through characterization of the C_28_-diol product by GC-MS/^1^H-NMR analysis, as reported for studies on *Tb* SMT1 and *Ac* SMT2 (Appendix A), [20,28,29]. Importantly, the C_28_-diol product is not a biosynthesis product but a saponification product; its identification serves as a chemical signature for the compound in its biosynthetic state as a C_28_-intermediate-SMT adduct, reported first in studies on the yeast SMT incubated with the suicide substrate 26,27-dehydrozymosterol [30,31], then for the natural plant substrate 24-methyl cycloartenol tested against the soybean SMT1 [44,45]. An issue now is whether fungal sterols (such as CHT), which can bind irreversibly to *Tb* SMT [29] could behave as growth inhibitors against the kinetoplastid parasite in vitro and in vivo.

In our testing of CHT and ERGT against *T. brucei* we observed that they are trypanocidal at nanomolar concentrations, consistent with their predicted suicide inhibitor/antimetabolite effect on ergosterol biosynthesis. Consequently, we desired to explore their efficacy in a mouse model of infection. Previous efforts to identify more selective irreversible inhibitors for this enzyme continued and led to the discovery of pro-drugs and their associated zymosterol derivatives [27,34,44] that can serve as suicide inhibitors of ergosterol biosynthesis [33,46]. The EC_50_ of these compounds were in the range of 10–20 µM with specificity indexes 3–6 [34,46]. In contrast to synthetic 26,27-dehydrolanosterol or 26-fluorolanosterol [34,46], the EC_50_s of natural CHT and ERGT against the *T. brucei* bloodstream form reported here are 2.9 ± 0.2 nM and 0.52 ± 0.05 µM, which are much lower than the previously identified suicide inhibitors. CHT is more potent than ERGT. This is possibly because of differences in productive binding to the *Tb* SMT, which could be due to the distinct binding sites for the two inhibitors and/or affected by the expression level of *Tb* SMT, sensitive to pressures of lipoprotein cholesterol. In any case, the selectivity index of these compounds for the BF was about three orders of magnitude as determined in vitro, showing that the compound is non-toxic to the parasite host. In addition, these antimetabolites showed a long-lasting effect for the bloodstream form, showing irreversible inactivation of the target enzyme, 24-SMT.

In animal models for *T. brucei* acute infection, CHT/ERGT showed promising results. CHT/ERGT-treated mice survived longer than the untreated control group. Although the parasite counts in the blood at the initial 5–6 days of infection were below the detection levels, once they cross this threshold they increase exponentially. There are multiple possibilities for these results. (1) *T. brucei* could be adapted by changing the means of drug uptake/efflux at the latter days of treatment or became resistant by other means. (2) It is also possible that the parasites were cleared from the bloodstream during initial treatment; however, a population that hides inside tissues (adipose tissue or brain) escaped, because drug concentration could be below the cidal levels in these tissues and that creates a space for their adaptation. These adapted parasites emerged in larger numbers at later time points. (3) The third possibility is that the antimetabolites are metabolized in the host and that metabolism was induced with a longer treatment period, thus the serum levels of these compounds dropped at the latter days and parasite population increased exponentially. We didn’t find any toxicity of CHT/ERGT at the dosage of treatment. We also didn’t observe any better effect by increasing the dose from 50 to 100 mg/kg/day via intraperitoneal injection. It was also not recommended for more frequent administration of these inhibitors due to the smaller size of the animals and volumes required to administer. However, these results warrant further trials with larger animals and more frequent administration of these effective compounds and perhaps in combination therapy with conventional antitrypanosomal drugs. Overall, we found that CHT and ERGT are promising candidates to develop drugs targeting 24-SMT in both *A. castelanii* and *T. brucei* that cause deadly diseases in humans.

## 4. Material and Methods

### 4.1. Tb Cell Culture

*T. brucei* strain Lister 427 BF cells (ATCC NR-42009) were cultured in HMI-9 medium supplemented with 10% heat-inactivated fetal bovine serum (Bio-Techne, Minneapolis, MN, USA) and 10% Serum Plus (Thermo Fisher Scientific, Waltham, MA, USA) in a CO_2_ incubator (5% saturation) at 37 °C as described [47]. To measure cell growth, BF cells were inoculated at 10^4^/mL or as indicated, the parasite numbers were counted in a hemocytometer chamber using a phase-contrast microscope and plotted versus time of incubation.

### 4.2. Preparation of ERGT and CHT for Treatment and Sterol Analytics 

Cholesta-5,7,22,24-tetraenol (CHT) and ergosta-5,7,22,24(28)-tetraenol (ERGT) were purified from the yeast mutants ERG6 and ERG5, respectively [28]. Stock solutions of ERGT and CHT were made in DMSO at a concentration of 10 mM, diluted accordingly before addition to the culture medium. For in vivo experiments, the stock solution of ERGT and CHT was made in DMSO at 100 mg/mL. At the time of treatment stock solution was diluted 25-fold in 1X PBS and injected intraperitoneally at the dose of 50 mg/kg or 100 mg/kg body weight for each mouse.

### 4.3. Inhibition Studies in Culture

Determination of EC_50_/EC_90_ and wash out experiments to assess the effect of sterol biosynthesis inhibitor on cell growth in vitro were performed against cultured *T. brucei* cells as described previously [24,28]. Briefly, for determination of EC_50_/EC_90_, BF cells from a logarithmic phase culture were inoculated at 10^4^/mL in 24-well plates. CHT/ERGT were serially diluted in medium from the stock solution as described above. Triplicate wells were set up for each concentration. Cells were counted each day for 2 days from each well twice and plotted against time with calculated standard deviations. For washout studies, the BF cells were pretreated with CHT and ERGT at 5 × EC_50_ concentrations for 2 h. Cells were harvested, washed to remove drugs, and reinoculated in fresh medium at different cell numbers (10^4^/mL, 10^3^/mL, 10^2^/mL and 10^1^/mL). The untreated control cells were run in parallel starting with different inoculum sizes. Cell numbers were counted each day for 4 days and plotted against time in culture.

### 4.4. Giemsa Staining

*T. brucei* BF was grown in appropriate medium in the presence or absence of the drugs. At different time points cells were harvested, resuspended in fresh medium at a concentration of 10^5^–10^6^/mL, spread on a slide and left to air dry. Slides were flooded with 10% Giemsa stain solution and kept at room temperature for 20–30 min. Slides were washed with water to remove the excess dye and allowed to dry. Image was taken with a Keyence phase-contrast microscope using 40× objective

### 4.5. Electron Microscopy 

Cells were fixed in 2% (*v*/*v*) glutaraldehyde and 2% (*w*/*v*) paraformaldehyde in 0.1 M sodium cacodylate buffer (SCB), pH 7.2 [48]. Cells were then washed with SCB, post-fixed with 1% osmium tetroxide in SCB, stained with 0.5% aqueous magnesium uranyl acetate, dehydrated, and embedded in Spurr’s resin. Blocks were sectioned at 50–70 nm thickness and stained with 5% (*w*/*v*) uranyl acetate in 1% acetic acid and 0.4% (*w*/*v*) lead citrate in 0.1 N NaOH. Section grids were inserted into FEI CM12 twin lens 420 transmission electron microscopes (FEI) to capture images.

### 4.6. Animal Experiments

In the model of acute infection (Haubrich et al., 2015), female Balb/C mice (6–8 weeks old) for *T. brucei* infection were purchased from the Envigo laboratory, weight 25 g on average: 5 mice/group. Animals were allowed to be acclimated for 3–5 days in the Animal Care facility in Meharry Medical College. For *T. brucei* infection, BF cells were harvested from exponentially growing cultures and resuspended in 1X PBSG (phosphate buffered saline containing 5 mM glucose) at a concentration of 2 × 10^4^/mL. Each mouse was injected intraperitoneally with 0.5 mL of cell suspension (1 × 10^4^ cells). In these conditions, parasitemia against Tb reaches maximum of 1 × 10^9^ cells on day 6 after infection. Mice were treated separately with ERGT and CHT (50 or 100 mg/kg body weight/mice/day) via intraperitoneal injection starting at the day of infection. The control mouse group received only vehicle. Parasitemia was monitored daily by counting the parasite numbers in blood (3–5 µL) collected by tail snipping. To reduce the pain and distress and to set up a humane end-point, mice were sacrificed when the parasitemia levels reached >5 × 10^8^ cells/mL of blood and death is considered one day after euthanasia. All mice were closely monitored for any distress or pain by periodical assessment of body weight, food intake, hair coat, and activities in accordance with the animal protocol guidelines of the Meharry Medical College. Mice under severe distress were sacrificed after consultation with the Animal Care Facility Veterinarians. All studies were conducted in accordance with National Institutes of Health guidelines for the use of experimental animals, and the protocols were approved by the Meharry Institutional Animal Care and Use Committee (protocol number: 141017MC172, Animal Welfare assurance number: A3420-01)

## 5. Conclusions

We identified two natural byproducts of yeast sterol metabolism with potent anti-trypanocidal property but non-toxic to human cells. We believe these studies bridged, for the first time, the natural products involving sterols other than cholesterol and their chemical biology with the health sciences. We showed that CHT/ERGT, which can act as substrates of *Tb* SMT, and form an adduct with the enzyme during catalysis, can carry out their inhibitory effects on ergosterol biosynthesis with lethal consequences. Importantly, these antimetabolites work at nanomolar concentrations in vitro to inhibit cell growth. These products are also effective in vivo in a mouse model of *T. brucei* acute infection, doubling the survival time of the animals. Therefore, these potent trypanocidals deserve further investigation to optimize their treatment doses and pharmacokinetics to develop novel chemotherapies for AT.

## Figures and Tables

**Figure 1 molecules-27-04088-f001:**
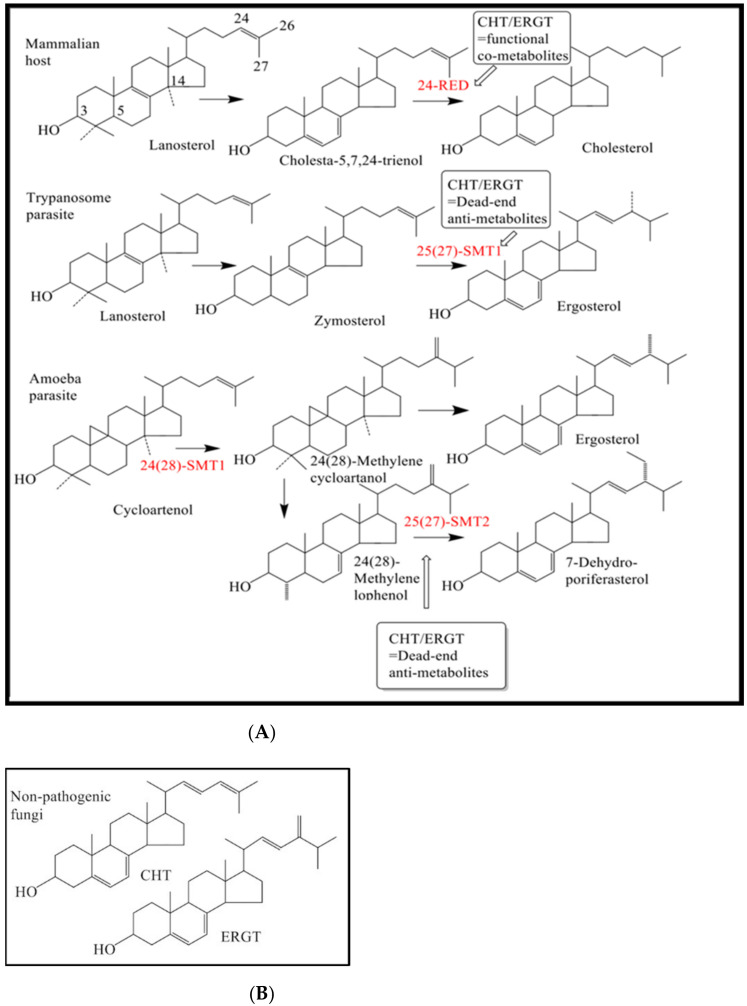
**Differences in inhibition of steroidogenesis affected by steroidal antimetabolites.** As shown, in ergosterol synthesis CHT and ERGT specifically inhibit the D25(27)-SMT in *Tb* and *Ac* via suicide (irreversible) inhibition. Alternatively, in cholesterogenesis neither CHT nor ERGT attach irreversibly to any enzyme in the pathway (**A**). Antimetabolites ERGT and CHT derived from yeast (**B**).

**Figure 2 molecules-27-04088-f002:**
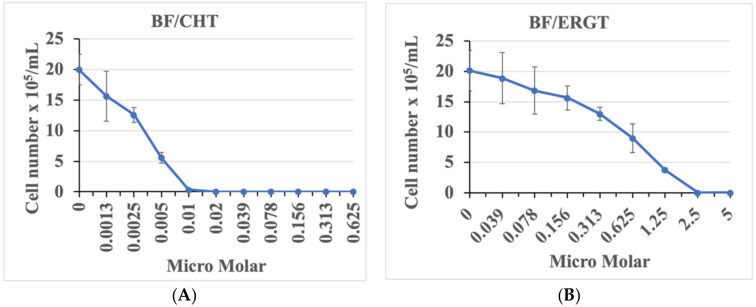
***T. brucei* cell growth at different concentrations of CHT and ERGT.***T. brucei* BF cells were inoculated at a cell density of 10^4^/mL in HMI-9 medium containing different concentrations of CHT (0–0.62 µM) (**A**) and ERGT (0–5 µM) (**B**). Cells were counted after 2 days. Cell numbers were plotted against the concentrations of drugs to determine the EC_50_ of each compound for each type of cell. Experiments were done in triplicate to calculate the standard deviations.

**Figure 3 molecules-27-04088-f003:**
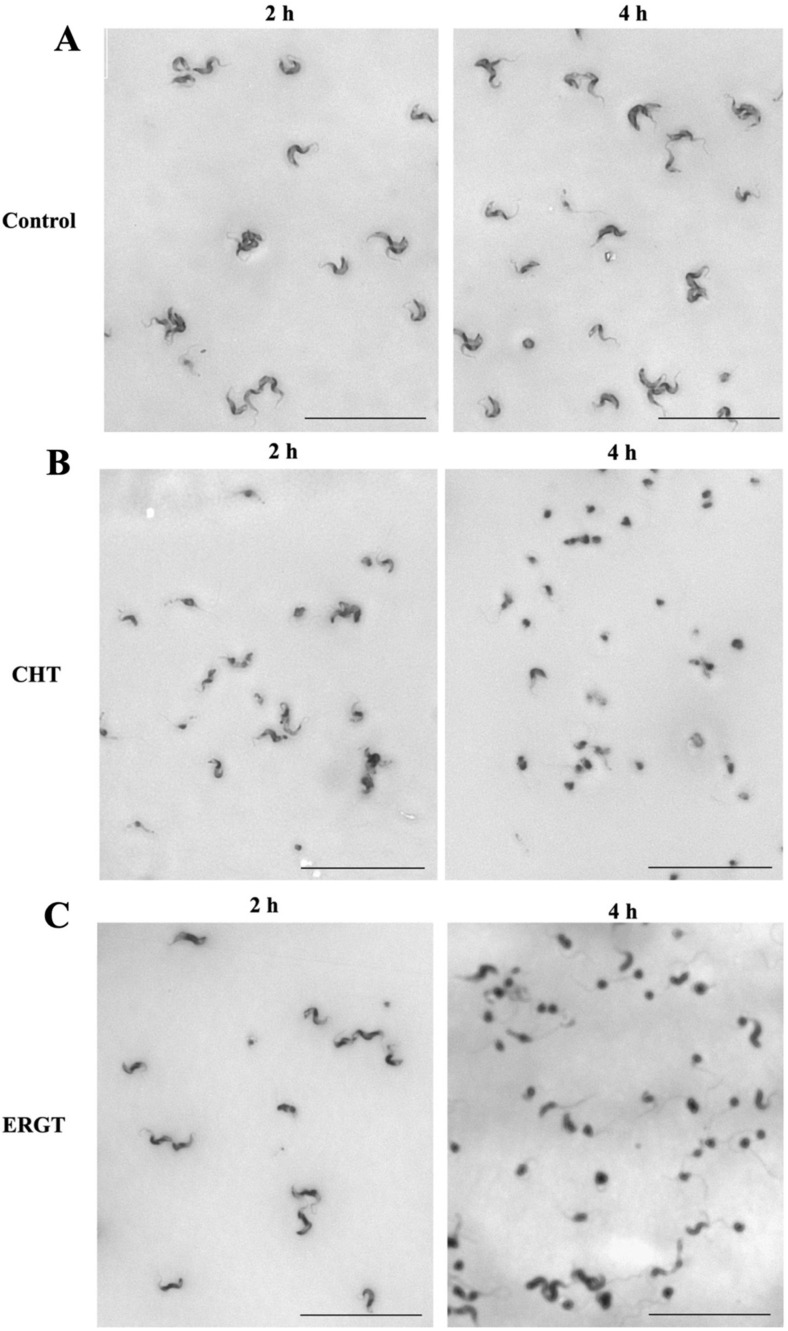
**Effect of CHT/ERGT treatment on cell morphology.** *T. brucei* BF cells (1 × 10^6^/mL) were incubated with CHT and ERGT at their respective 5 × EC_50_ concentrations. Untreated control cells were used in parallel (**A**). At different time points (2 and 4 h) cells (100 µL of the culture) were harvested, spread on microscope slides, and stained with Giemsa solution. Images were taken with a Keyence phase-contrast microscope using 40× objective. A major change in morphology was observed for cells treated with CHT (**B**) and ERGT (**C**) at 4 h time point. Experiments were repeated three times and multiple areas on the slides were observed for each sample. Scale bars represent 50 µm.

**Figure 4 molecules-27-04088-f004:**
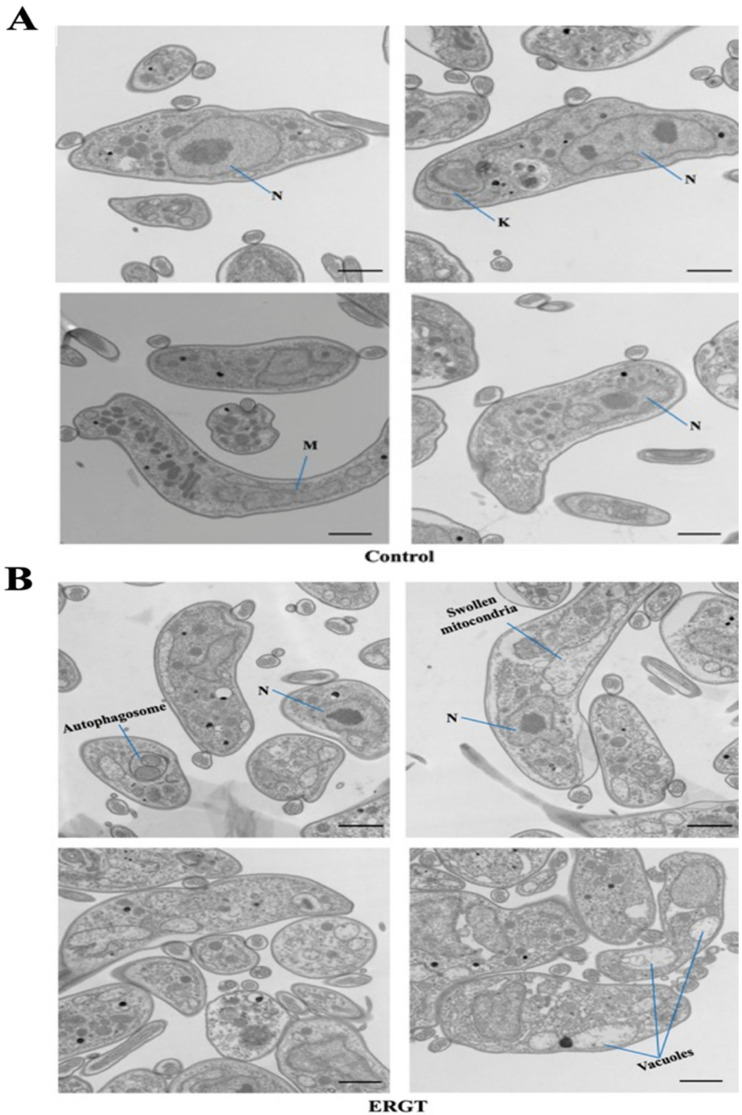
**Electron microscopy of the ERGT-treated *T. brucei.*** *T. brucei* BF cells were incubated with ERGT at 5 × EC_50_ for 4 h. Cells were harvested, treated with fixative solution, and prepared for electron microscopy as described in the materials and methods. Untreated control cells were run in parallel (**A**). Images from cells treated with ERGT are shown in (**B**). N and K represent nucleus and kinetoplast. M represents mitochondria. Presence of autophagosome, swollen mitochondria, and large vacuoles found in treated cells are indicated. Two individual biological replicates were used for EM and multiple sections from each sample were examined. Scale bars represent 1 µm.

**Figure 5 molecules-27-04088-f005:**
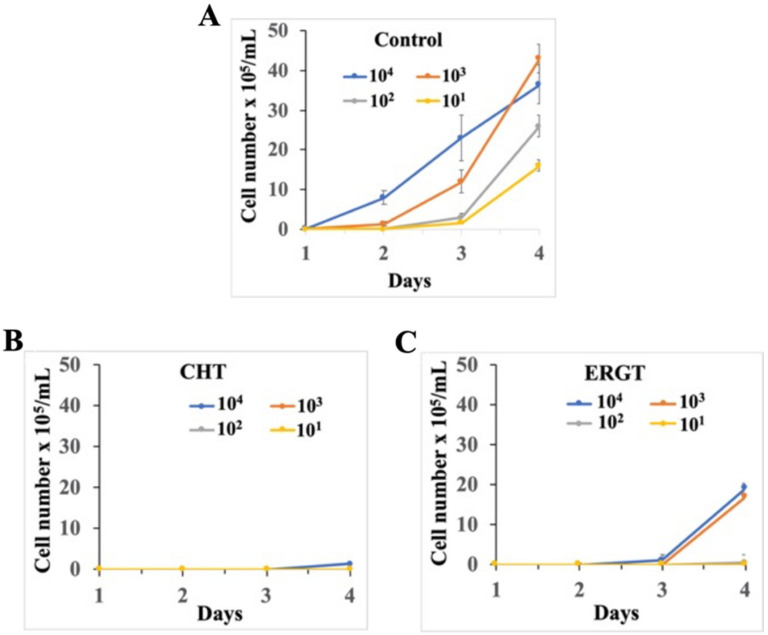
**Extended effect of CHT and ERGT on *T. brucei* BF cell growth.** The BF cells were pretreated with CHT and EREGT at 5 × EC_50_ concentrations for 2 h. Cells were harvested, washed to remove drugs and reinoculated in fresh medium at different cell numbers (10^4^/mL, 10^3^/mL, 10^2^/mL and 10^1^/mL). The untreated control cells were run in parallel starting with different inoculum sizes. Cell number was counted each day for 4 days and plotted against time in culture. (**A**) control, (**B**) and (**C**) are CHT and ERGT treated cells, respectively. Standard errors were calculated from four independent experiments.

**Figure 6 molecules-27-04088-f006:**
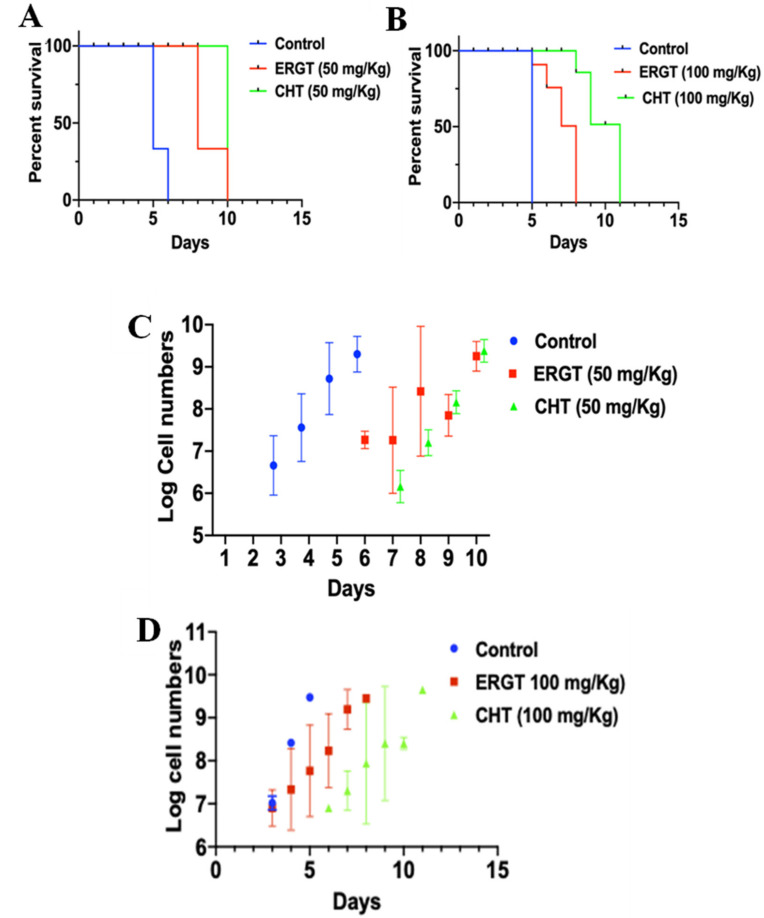
**Protective role of CHT/ERGT on animal infection with *T. brucei*.** Six groups of Balb/C mice (5 mice in each group) were injected intraperitoneally with *T. brucei* bloodstream form parasite (1 × 10^4^ cells/mouse). Two groups were treated with ERGT (50 mg/kg) or CHT (50 mg/kg) via intraperitoneal injection once per day post-infection (**A**). Another two groups of mice were treated with ERGT 100 mg/kg) or CHT (100 mg/kg) (**B**) via the same route once per day post-infection. Controls for these groups were injected with appropriate amount of buffer used to prepare the antimetabolites. The blood parasitemia levels were monitored each day post-infection by counting the number of *T. brucei* in blood collected from the tail vein. Survival curves for the control (infected but not treated with any drug) and infected mice treated with ERGT were plotted using Graphpad Prism. To reduce the pain and distress mice were sacrificed when the parasitemia levels reached >5 × 10^8^ cells/mL of blood and death is considered one day after euthanasia. (**C**,**D**) Log of parasite numbers in the blood of the infected mice are plotted against days post infection. Standard deviations were calculated from 5 counts from each mouse on each day. Parasite numbers on day 1 and 2 post-infection were below the detection limit.

## Data Availability

Not applicable.

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
