# Peer review of "Steroidal Antimetabolites Protect Mice against Trypanosoma brucei"

_molecules, 2022, doi:10.3390/molecules27134088_

Round 1

Reviewer 1 Report

The authors describe in this manuscript the in vitro and in vivo study of the anti-T. brucei potential of two steroidal inhibitors of sterol methyltransferase, CHT and ERGT. The work is interesting and well exposed; the organization of the article is clear and the subject fits well to the scope of the journal. Bibliographical references are suitable and in sufficient number to illustrate this work properly.

Several remarks should be brought to the attention of the authors:

- Line 25, rewording suggestion: replacing “anti-trypanosome” by “antitrypanosomal”.

- Lines 39, 40 and 430: please write « fexinidazole » and not « flexinidazole ».

- Line 40-41: the issue of resistance to fexinidazole is not trivial: are there no bibliographical references to illustrate it?

- Lines 57, 101, 291, 292, 351 and 402: please write “via” in italics.

- Line 66: A figure to recall the chemical structures of fexinidazole,  oxaboroles of interest previously mentioned, and eflornithine might be welcome here. Concerning eflornithine, since the covalent nature of its mechanism is of interest here, the molecular aspects of its irreversible interaction with ornithine decarboxylase could be illustrated here.

- Lines 81-84: please check the grammar of this sentence.

- Lines 83-84: please write Acanthamoeba in italics.

- Line 87, rewording suggestion: replacing “microbes” by “protozoa” or “protozoans”.

- Line 98: please consider reworking Figure 1 by spacing the molecules from each other to make it more readable, even if it means making the figure wider and higher (also in order to correctly integrate the bottom box, which goes beyond the frame). Besides, the name and structure of CHT and ERGT should be better highlighted as these 2 molecules are the core of this work.

- Line 104: from a chemical point of view, SMT is not "complexed" by its inhibitor. Please consider replacing the word “complex” by “inhibit” or “inactivate”.

- Line 105: please correct the “observed s pair” part of the sentence.

- Line 119: please be consistent in all occurrences of TbSMT, as the Tb should effectively be written in italics.

- Lines 125-130: The positioning of Figure 2 is not correct because it includes these results in the introduction. Please move this figure before paragraph 2.2.

- Line 129, 181, 191, 205, 220, 246, 325 and 327: please write “EC50” with the number in index.

- Line 141: same remark for “AcSMT1” as for TbSMT at line 119.

- Line 159-160 and 168-169: please write “EC50” and “EC90” with the numbers in index.

- Line 159: please add a space between “2.9” and “±”.

- Line 169, 180, 236, 290 and 298: please use the correct symbol “×” and not “x” for multiplicative factors.

- Line 177: please write “very” and not “vary”.

- Line 191: please remove the capital letter from “Electron”, as it is neither a proper noun, nor the first word of a sentence.

- Line 207: please write “Images” with a capital letter.

- Line 210: the fact that additional pictures are included in the supplement is not correct, as only Figure S1 is available in supplementary materials. Extra electron microscopy images would be welcome as supplementary data, however.

- Line 216, rewording suggestion: the word “complex” could be replaced by “adduct”, which is more accurate from a chemical point of view.

- Line 256: is there not a word missing between “Tb mouse” and “of infection”?

- Line 266: maybe “were died” should be considered for rewording.

- Line 278: please write “stayed alive” and not “stay alive” for concordance of the tenses.

- Line 310: please write “1H NMR” with the 1 in superscript.

- Line 328: please add a space between “2.9” and ±”.

- Line 356: please write “antitrypanosomal” in a single word, as there is no reason to use a hyphen.

- Line 363: please write “37 °C with the proper “°” symbol.

- Line 370: please write “in vivo” in italics.

- Line 375: please write “EC50” and “EC90” with the numbers in index.

- Paragraph 4.3: although already described, a brief report of the inhibition studies protocol would be valuable here.

- Line 394: please specify the inbred model of mice that was used.

- References: bibliographic references concerned need to be reformatted to show DOI, according to the citation style of the Multidisciplinary Digital Publishing Institute. See the publisher's website.

Author Response

Responses to Reviewer 1

Thank you for a thorough review of our manuscript. We have incorporated all the comments to edit our manuscript (a track change file is attached). Our responses to few major points are as follows.

Comment #1: Line 66: A figure to recall the chemical structures of fexinidazole, oxaboroles of interest previously mentioned, and eflornithine might be welcome here. Concerning eflornithine, since the covalent nature of its mechanism is of interest here, the molecular aspects of its irreversible interaction with ornithine decarboxylase could be illustrated here.

Response: This is a good idea; however, we would like to reserve this for a review article. The chemistry of protein inactivity resulting from the suicide substrate is well known and documented in the references cited in our paper.

Comment #2: Line 98: please consider reworking Figure 1 by spacing the molecules from each other to make it more readable, even if it means making the figure wider and higher (also in order to correctly integrate the bottom box, which goes beyond the frame). Besides, the name and structure of CHT and ERGT should be better highlighted as these 2 molecules are the core of this work.

Response: We have now separated the anti-metabolites (the inset box) as Figure 1B. We spaced the chemical structure as much as possible to keep the figure within one page.

Comment #3: Line 104: from a chemical point of view, SMT is not "complexed" by its inhibitor. Please consider replacing the word “complex” by “inhibit” or “inactivate”.

Response: Suicide inhibitors form covalent bond with SMT to inactivate this enzyme, which means that it forms an irreversible complex with the enzyme. We used the term adduct in some cases.

Comment #4: Paragraph 4.3: although already described, a brief report of the inhibition studies protocol would be valuable here.

Response: We have elaborated this method

Comment #5: References: bibliographic references concerned need to be reformatted to show DOI, according to the citation style of the Multidisciplinary Digital Publishing Institute. See the publisher's website.

Response: We think DOI will be added during reference check by the editorial office once the manuscript is approved for publication.

Reviewer 2 Report

The MS describes a new class of antibiotics, in form of fugal steroids against a protozoan Trypanosoma brucei infecting human and domestic animals. The parasite is transmitted via vectors, tsetse fly. In the study the effect of cholesta-5,7,22,24-tetraenol (CHT), and ergosta-5,7,22,24(28)-tetraenol (ERGT), was examined in vitro and in vivo was tested. CHT/ERGT inhibited cell growth in vitro yielding EC50 values in low nanomolar range; in vivo ERGT/CHT protected mice infected with T. brucei by doubling their survival time following daily treatment for 8-10 days. The study is well designed, methods are appropriately described, results are significant; however, some changes should be done.

Main points:

1.     Citing in text is not correct, must be quoted in square brackets. Change citing form in L377, 394.

2.     Incorrect figure citing. In text must be Figure rather than Fig. Figure 2 should be citied/mentioned before the figure presentation.

3.     It is clear why experiments were not standardized, i.e., a different number of repetitions was performed (Figure 2 three times, Figure 5 four experiments). What is the explanation?

4.     L133-151 previous studies are discussed more; therefore, this paragraph should be modified and moved to introduction section.

5.     L167-178 not correct text alignment, should be justified.

6.     I really suggest to include conclusions section. 

7.     REFERENCES should be corrected. Article titles should be written in unified style, Sentence case or Capitalize Each Word. Between pages should be long dashes.

Other points:

1.     L38 Drug for Neglected Tropical Disease Initiative shouldn’t it be DNTDi rather than DNDi?

2.     L64 WHO, you have already used abbreviation above

3.     L109, L137 in vitro in vivo should be italic

4.     L113-114 “and references cited therein” should be italic

5.     Explanation what is BF is in L361, although the abbreviation BF is mentioned for the first time in L168

6.     Figure 3. Scale bars are missing in B panel

7.     L192 delete unneeded dot after (EM).

8.     L194 replace to “…(K) and mitochondria in trypanosomes….”

9.     L198 change to “in the control (Fig. 4A)...any cristate structures (Fig. 4B)”.

10.  Figure 4. Nucleus also should be marked in B panel top left figure

11.   Figure 6. Figure and legend should be in the same page.

12.  L301 please insert (~103/ml)

13.  L413-415 please correct authors’ contribution according to form. “Conceptualization, data curation, formal analysis, investigation, methodology, software, supervision, validation, visualization, writing—original draft preparation, writing-review and editing”

Author Response

Responses to Reviewer 2

Comment #1: Citing in text is not correct, must be quoted in square brackets. Change citing form in L377, 394.

Response: Corrected

Comment #2: Incorrect figure citing. In text must be Figure rather than Fig. Figure 2 should be citied/mentioned before the figure presentation.

Response: Done

Comment #3:    It is clear why experiments were not standardized, i.e., a different number of repetitions was performed (Figure 2 three times, Figure 5 four experiments). What is the explanation?

Response: All experiments were performed at least 3 times. If the results are variable, we repeated more. All methods used in this manuscript are straightforward and well-standardized in the lab.

Comment #4: L133-151 previous studies are discussed more; therefore, this paragraph should be modified and moved to introduction section.

Response: This paragraph servs as the rationale to test the antimetabolites against Tb cell growth both in vitro and in vivo. Therefore, we would like to keep this at the beginning of the result section.

Comment #5:   L167-178 not correct text alignment, should be justified.

Response: Corrected

Comment #6:   I really suggest to include conclusions section

Response: Conclusion has been added

Comment #7:   REFERENCES should be corrected. Article titles should be written in unified style, Sentence case or Capitalize Each Word. Between pages should be long dashes.

Response: We checked the reference style and made necessary correction.

Comment #8: L38 Drug for Neglected Tropical Disease Initiative shouldn’t it be DNTDi rather than DNDi?

Response: DNDi is correct. It represents Drug for neglected Disease Initiative. We have corrected the text

Comment #9: Figure 3. Scale bars are missing in B panel

Response: Corrected

Comment #10: L194 replace to “…(K) and mitochondria in trypanosomes

Response: The concatenated structure of the mitochondrial DNA is called kinetoplast-DNA or kDNA (K). We think the sentence is correct.

Comment # 11: Figure 4. Nucleus also should be marked in B panel top left figure

Response: Done

Comment # 12: Figure 6. Figure and legend should be in the same page.

Response: Adjusted.

Comment #13: L301 please insert (~103/ml)

Response: Done

Comment #13:  L413-415 please correct authors’ contribution according to form. “Conceptualization, data curation, formal analysis, investigation, methodology, software, supervision, validation, visualization, writing—original draft preparation, writing-review and editing”

Response: We have categorized author’s contribution accordingly

All other minor points have been corrected

Author Response

Responses to Reviewer 3

Comment #1: The language used, grammatical errors and sentence construction, make this already complex paper more difficult to follow and understand. This requires considerable attention before publication. (Please do not use contractions eg didn’t in formal scientific writing)

Response: We carefully checked for language and grammar throughout the manuscript.

Comment #2: Title: This title is too strong and overstates the findings. The survival time following infection was increased however the mice still succumbed to the disease, it also provided no indication of the in vitro studies performed.

Response: We understand the point. However, we think the title represents the broader outcomes that bridge natural products involving sterols for the first time and their chemical biology affecting the health sciences. Since other reviewers did not raise any concern, we would like to keep the title as it is.

Comment #3: Introduction • Change ‘flexinidazole’ to fexinidazole throughout (including references)

  • Provide a reference supporting nifurtimox / fexinidazole cross resistance
  • Vaniqua is the DFMO/eflornithine tradename for the formulation used as a hair depilatory cream, Ornidyl is the 22mg/ml solution produced by Aventis for treatment of human African trypanosomiasis.
  • Eflornithine is not effective against infections caused by T. b. rhodesiense and does have recognised adverse effects

Response:

  • Spelling has been corrected.
  • Nifurtimox and fexinidazole both are nitroimidazole derivatives that requires nitro reductase (NTR) for their activation. It has been shown in laboratory that mutation of NTR gene can cause resistance to both drugs [Wyllie et al., J. antimicrobial Chemotherapy, 2016,71,625-634]. We have included this reference in the manuscript and modified the sentence in the Introduction stating that ‘fexinidazole has potential to develop cross-resistance with nifurtimox’.
  • We have changed the trade name for eflornithine as Ornidyl and revised the statement.

Comment # 4:  Figure 1; the pathways need to be better spaced and described in the legend. Inclusion of the antimetabolites as a separate figure or panel may be beneficial.

Response: We have now separated the anti-metabolites (the inset box) as Figure 1B. We spaced the chemical structure as much as possible to keep the figure within one page.

Commment # 5: Results

  • Move figure 2 further into the results section. Add EC50/EC90 details to the plots.
  • There are large sections of text included in the ‘results’ that are better suited to the introduction, materials and methods or discussion section. Only the experimental results obtained in the current study, or those with direct bearing on the findings, should be described in this section of the paper.

Response:

  • The position of figure 2 adjusted. We kept the grid lines in the graphs for quick estimation of the EC50 and EC90
  • Please see our responses to reviewer 2. We believe these segments are needed for the clarity and rationale of the experiments

Comment # 6: Line 145: change ‘Ac trophophyte cells’ to Ac trophozoites

Response: Corrected

Comment #7: Section 2.2; Why was 5xEC50 chosen for investigating effect of antimetabolites on in vitro trypanosomes? This is above EC90 in both cases. Please provide explanation in the Materials and Methods section.

Response: We treated cells at 5xEC50 for a shorter period (2h) to see the biological effect. EC50 was determined after treating cells for 48 h. We selected this concentration and time because at this condition cell motility and viability were minimally affected. As requested, we added further clarification in the result. This is a reasonable technique used for drug inhibition studies.

Comment # 8: Inclusion of a trypan blue viability assay would have been beneficial here. Giemsa provides some information re morphology but little on viability.

Response: We used Giemsa to investigate the changes in the morphology and not for viability. Cell viability was measured by counting the motile cells by hemocytometer under microscope

Comment # 9: Line 199: Normal mitochondria in trypanosomes lack cristae, this is not an effect of the treatment.

Response: There are plenty of evidence that the single mitochondrion in T. brucei has cristae. The procyclic form has more than the bloodstream form, but both forms have cristae. The reviewer may see the following reference paper and papers published by the first author of this manuscript.

  • Bily T, Sheikh S, Mallet A, Bastin P, Perez-Moga D, Lukes J, Hashimi H. 2021. Ultrastructureal changes of the mitochondrion during the life cycle of Trypanosoma brucei. J. Eukaryot Microbiol 68:e12846

Comment # 10: Were any statistical analyses performed to determine whether the differences seen between the two anitmetabolites were significant? Discussion

Response:  CHT and ERGT showed differences in their trypanocidal property at least in vitro. We think these different antimetabolite treatment outcomes are to be expected since the ERGT-CHT inhibited growth response is subject to endogenous TbSMT1 acceptance of the fungal sterols.  As reported, a first methylation substrate (e.g., CHT) is converted by the Tb enzyme about ten times more effectively than a second methylation substrate (e.g., ERGT) [Zhou et al., 2006]. We discussed this in the manuscript. We didn’t compare the effect of CHT and ERGT with statistical significance.

Comment # 11: Line 341; The ‘adaption’ of parasites residing in the body tissues seems unlikely. Adaption would need to be rapid as the infection is only sustained for approximately 10-days in treated mice.

  • This also applies re ‘tissue’ resident population Materials and Methods
  • Additional methodological details are required in this section.

Response:

  • We agree with the reviewer. However, trypanosome metabolic patterns are very flexible, and they are capable to adapt in widely different environments very rapidly (within hours or minutes). Therefore, we think that although unlikely, it is a possible scenario.
  • In this manuscript we just monitor the parasite levels in the blood. Therefore, any method for tissue-resident parasites is irrelevant

Comment #12: Please expand 4.1 – A brief description of the procedures used is required in addition to the reference. Eg Were the parasites cultured in plates or flasks? What was seeding density?

  • Please expand 4.3. As above
  • Section 4.6; What mouse strain did you use? Death is not an ethically acceptable endpoint for animal work. Were humane endpoints set? You say body weight was assessed but no data is given. Parasitemia was measured daily but was this used to direct euthanasia?

Reference format and other minor points have been corrected appropriately

Response: We included the information requested. It is mentioned in the legend of Figure 6 that “To reduce the pain and distress mice were sacrificed when the parasitemia levels reached > 5 X 108 cells/ml of blood and death is considered one day after euthanasia”. Now added in the method section. There was no change in body weight during the experimental period, therefore the results were not shown.

All other minor points have been corrected.